# Generalized $(\psi, \alpha, \beta)$ − Weak Contractions for Initial Value Problems

**Piyachat Borisut** [1] 📷, **Poom Kumam** [1,2,*] 📷, **Vishal Gupta** [3] 📷 **and Naveen Mani** [3] 📷

1   KMUTT-Fixed Point Research Laboratory, Room SCL 802 Fixed Point Laboratory, Science Laboratory Building, Department of Mathematics, Faculty of Science, King Mongkut's University of Technology Thonburi (KMUTT), 126 Pracha-Uthit Road, Bang Mod, Thrung Khru, Bangkok 10140, Thailand; piyachat.b@hotmail.com
2   KMUTT-Fixed Point Theory and Applications Research Group, Theoretical and Computational Science Center (TaCS), Science Laboratory Building, Faculty of Science, King Mongkut's University of Technology Thonburi (KMUTT), 126 Pracha-Uthit Road, Bang Mod, Thrung Khru, Bangkok 10140, Thailand
3   Department of Mathematics, Maharishi Markandeshwar (Deemed to be University), Mullana 133207, Haryana, India; vishal.gmn@gmail.com (V.G.); naveenmani81@gmail.com (N.M.)
*   Correspondences: poom.kum@kmutt.ac.th

**Abstract:** A class of generalized $(\psi, \alpha, \beta)$− weak contraction is introduced and some fixed-point theorems in a framework of partially ordered metric spaces are proved. The main result of this paper is applied to a first-order ordinary differential equation to find its solution.

**Keywords:** coincidence point; fixed point; generalized $(\psi, \alpha, \beta)$− weak contraction; partially ordered metric space; initial value problem

---

## 1. Introduction

The Banach contraction principle is a fundamental result in fixed-point theory Banach contraction principle [1]. This principle has been generalized in different directions by various researchers because of its usability and applicability.

In 1973, Geraghty [2] defined a class of functions $\alpha$ as follows:

**Definition 1.** *[2] Define $F = \{\alpha | \alpha : [0, \infty) \to [0, 1)\}$ which satisfies the condition*

$$\alpha(t_n) \to 1 \quad implies \quad t_n \to 0.$$

Geraghty [2] investigated the following theorem, which is known as Geraghty contraction.

**Theorem 1.** *[2] Let $(X, d)$ be a complete metric space and let $f : X \to X$ be a map. Suppose there exists $\alpha \in F$ such that for each $x, y \in X$*

$$d(fx, fy) \le \alpha(d(x, y))d(x, y).$$

*Then $f$ has a unique fixed point $z \in X$.*

**Definition 2.** *[3] Let $\Psi$ denote the class of function $\psi : [0, \infty) \to [0, \infty)$ which satisfies the following conditions:*

*(i)   $\psi$ is continuous and non-decreasing,*
*(ii)   $\psi(t) = 0$ if and only if $t = 0$.*

In 1997, Alber and Guerre-Delabriere [4] suggested a generalization of Banach contraction mapping by introducing the concept of $\phi$ -weak contraction in Hilbert space. Rhoades [5] showed that the result of Alber and Guerre-Delabriere [4] is still valid in complete metric spaces.

**Definition 3.** *[5] A self map T is said to be weakly contractive map if there exist a function $\phi : [0, +\infty) \to [0, +\infty)$ such that $\phi$ is continuous, non-decreasing and $\phi(t) = 0$ if and only if $t = 0$ and satisfying*

$$d\left(Tx, Ty\right) \leq d\left(x, y\right) - \phi(d\left(x, y\right))$$

*for all $x, y \in X$.*

**Theorem 2.** *[5] Let $(X, d)$ be a complete metric space and T be a weakly contractive self map on X. Then T has a unique fixed point in X.*

**Remark 1.** *Rhoades [5] observed that every contraction map T on X with contractive constant k is a weakly contractive map with $\phi(t) = (1 - k)t, t > 0$. However, its converse is not true.*

In 2008, Dutta and Choudhury [6] gave a generalization of weakly contractive mapping by defining $(\psi, \phi)$ -weak contraction in complete metric spaces.

**Definition 4.** *[6] Self map T is said to be $(\psi, \phi)$ weak contraction, if for each $x, y \in X$,*

$$\psi(d(Tx, Ty)) \leq \psi(d(x, y)) - \phi(d(x, y)),$$

*where $\psi, \phi : [0, \infty) \to [0, \infty)$ are both continuous and monotone non-decreasing functions with $\psi(t) = 0 = \phi(t)$ if and only if $t = 0$.*

**Theorem 3.** *[6] Let $(X, d)$ be a complete metric spaces and self map T be a $(\psi, \phi)$ weak contraction. Then T has a unique fixed point.*

Zhang and Song [7] defined and introduced a proper extension of $\phi-$ weak contraction namely, generalized $\phi-$ weak contraction.

**Definition 5.** *[7] Self maps T and R are said to be generalized $\phi-$ weakly contractive maps if there exist a function $\phi : [0, +\infty) \to [0, +\infty)$ such that $\phi$ is continuous, non-decreasing and $\phi(t) = 0$ if and only if $t = 0$ and satisfying*

$$d\left(Tx, Ry\right) \leq M\left(x, y\right) - \phi(M\left(x, y\right)),$$

*where,*

$$M(x, y) = \max\left\{d(x, y), d(x, Tx), d(y, Ry), \frac{[d(y, Tx) + d(x, Ry)]}{2}\right\}$$

*for all $x, y \in X$.*

**Theorem 4.** *[7] Let $(X, d)$ be a complete metric space and T and R are generalized $\phi-$ weakly contractive self maps on X. Then T and R have a unique common fixed point in X.*

Doric [8] extended the result of Zhang and Song [7] by defining generalized $(\psi, \phi)-$ weak contraction and proved some fixed-point theorems.

**Definition 6.** *[8] Self maps T and R are said to be generalized $(\psi, \phi)-$ weakly contractive maps if it satisfies*

$$\psi(d\left(Tx, Ry\right)) \leq \psi(M\left(x, y\right)) - \phi(M\left(x, y\right)),$$

*for all $x, y \in X$, where $\psi : [0, +\infty) \to [0, +\infty)$ such that $\psi$ is continuous, non-decreasing and $\psi(t) = 0$ if and only if $t = 0$, $\phi : [0, +\infty) \to [0, +\infty)$ such that $\phi$ lower semi-continuous function and $\phi(t) = 0$ if and only if $t = 0$ and*

$$M(x, y) = \max \left\{ d(x, y), d(x, Tx), d(y, Ry), \frac{[d(y, Tx) + d(x, Ry)]}{2} \right\}$$

**Theorem 5.** *[8] Let $(X, d)$ be a complete metric space and T and R are generalized $(\psi, \phi)-$ weakly contractive maps on X. Then T and R have a unique common fixed point in X.*

The existence of a fixed point for contraction mappings in partially ordered metric spaces was considered initially by Ran and Reurings [9]. In 2008, Agarwal et al. [10] extended the results of Ran and Reurings [9] for the case of generalized $\phi$-contractions as follows:

**Theorem 6.** *[10] Let $(X, \preceq)$ be a partially ordered set, and suppose that there exists a metric $d \in X$ such that $(X, d)$ is a complete metric space. Let $T : X \to X$ be an increasing operator such that the following three assertions hold:*

*(i)* *there exists an increasing mapping $\phi : R_+ \to R_+$ with $\lim_{n \to \infty} \phi^n(t) = 0$ for each $t > 0$, such that for each $x, y \in X$ with $x \succeq y$ we have*

$$d(Tx, Ty) \leq \phi \left( \max \left\{ d(x, y), d(x, Tx), d(y, Ty), \frac{1}{2}[d(x, Ty) + d(y, Tx)] \right\} \right),$$

*(ii)* *there exists $x_0 \in X$ with $x_0 \preceq Tx_0$,*
*(iii)* *T is continuous or if an increasing sequence $x_n \subset X$ converges to $x \in X$, then $x_n \preceq x$ for all $n \in N$.*

*Then T has at least one fixed point in X.*

In 2009, Harjani and Sadarangni [11] proved some fixed-point theorems as a version of Rhoades [5] and Dutta and Choudhury [6] for weakly contractive mappings in ordered metric spaces.

**Theorem 7.** *[11] Let $(X, \preceq)$ be a partially ordered set, and suppose that there exists a metric $d \in X$ such that $(X, d)$ is a complete metric space. Let $T : X \to X$ be a continuous and non-decreasing mapping such that*

$$d(Tx, Ty) \leq d(x, y) - \phi(d(x, y)) \quad \text{for all } x \geq y,$$

*where $\phi : [0, \infty) \to [0, \infty)$ is continuous and non-decreasing function such that $\phi$ is positive in $(0, \infty)$, $\phi(0) = 0$ and $\lim_{t \to \infty} \phi(t) = \infty$. If there exists $x_0 \in X$ with $x_0 \preceq Tx_0$, then T has a fixed point.*

**Theorem 8.** *[12] Let $(X, \preceq)$ be a partially ordered set, and suppose that there exists a metric $d \in X$ such that $(X, d)$ is a complete metric space. Let $T : X \to X$ be a continuous and non-decreasing mapping such that*

$$\psi(d(Tx, Ty)) \leq \psi(d(x, y)) - \phi(d(x, y)) \quad \text{for all } x \geq y,$$

*where $\psi$ and $\phi$ are altering distance functions. If there exists $x_0 \in X$ with $x_0 \preceq Tx_0$, then T has a fixed point.*

In 2010, Harandi and Emami [13] proved a version of Geraghty's result [2] in partially ordered metric spaces.

**Theorem 9.** *[13] Let $(X, \preceq)$ be a partially ordered set and suppose that there exists a metric $d \in X$ such that $(X, d)$ is a complete metric space. Let $T : X \to X$ be an increasing mapping such that there exists $x_0 \in X$ with $x_0 \preceq T(x_0)$. Suppose that there exits $\alpha \in F$ such that*

$$d(Tx, Ty) \leq \alpha(d(x, y))d(x, y) \quad for \ all \ x, y \in X \ with \ x \preceq y,$$

*and assume that either $T$ is continuous or $X$ is such that if there is an increasing sequence $\{x_n\} \to x$, $x \in X$, then $x_n \leq x$ for each $n \geq 1$. Also, if for all $x, y \in X$, there exists $z \in X$ which is comparable to $x$ and $y$. Then $T$ has a unique fixed point in $X$.*

In 2010, Altun and Simsek [14] introduced the notion of weakly increasing mappings and investigated some fixed-point results for non-decreasing and weakly increasing operators in a partially ordered metric space by using implicit relations. Singh (2015) [15] and He et al. (2017) [16] stated that a fixed-point theorem for generalized weak contractive map in a metric space is proven by generalizing some recent findings of Doric [8], Zhang and Song [7] .

**Definition 7.** *[14] Let $(X, \preceq)$ be a partially ordered set. Two mappings $T, R : X \to X$ are said to be weakly increasing if $Tx \preceq RTx$ and $Rx \preceq TRx$ for all $x \in X$.*

**Remark 2.** *Please note that two weakly increasing mappings need not be non-decreasing. Some examples are given in [14].*

**Definition 8.** *[17] Let $(X, d)$ be a metric space and $T, R : X \to X$ are given self mappings on X. The pair $(T, R)$ is said to be compatible if $\lim_{n \to \infty} d(TRx_n, RTx_n) = 0$, whenever $\{x_n\}$ is a sequence in X such that $\lim_{n \to \infty} Tx_n = \lim_{n \to \infty} Rx_n = t$ for some $t \in X$.*

In the following sections, we introduce and give an example of generalized $(\psi, \alpha, \beta)-$ weakly contractive maps and then prove some common fixed-point theorems in the sense of partially ordered complete metric space. For applicability and usability of our results in diverse areas, we give an application to find a common solution of Volterra-type integral equations.

## 2. Main Results

We begin with following definition.

**Definition 9.** *Three self maps $T, R, S$ are said to be a generalized $(\psi, \alpha, \beta)-$ weak contraction if for each $x, y \in X$*

$$\psi(d(Tx, Ry)) \leq \alpha(d(Sx, Sy))\beta(d(Sx, Sy)), \quad \forall \ x \geq y, \tag{1}$$

*where $\alpha \in F$, $\psi \in \Psi$ and $\beta : [0, \infty) \to [0, \infty)$ is a continuous function with condition*

$$0 < \beta(t) < \psi(t), \forall \ t > 0. \tag{2}$$

An example of generalized $(\psi, \alpha, \beta)-$ weak contraction is as follows:

**Example 1.** *Let $X = N \cup \{0\}$ . Define a metric*

$$d(x, y) = \begin{cases} x + y, & if \ x \neq y \\ 0, & if \ x = y. \end{cases}$$

*Then $(X, d)$ is a complete metric space. Consider three maps $T, R, S : X \to Q^+$ defined as*

$$Tx = \frac{x}{2}, \quad Rx = \frac{x}{3}, \quad Sx = x.$$

*Define maps $\psi, \beta : [0, \infty) \to [0, \infty)$ and $\alpha : [0, \infty) \to [0, 1)$ as $\psi(t) = 2t, \quad \beta(t) = t$ and $\alpha(t) = \frac{9}{10}$. Then clearly, three maps $T, R$ and $S$ are generalized $(\psi, \alpha, \beta)-$ weak contraction.*

Now we prove our main result.

**Theorem 10.** *Let $(X, \preceq)$ be a partially ordered set and assume that there exists a metric $d$ in $X$ such that $(X, d)$ is a complete metric space. Let $T, R, S : X \to X$ are a generalized $(\psi, \alpha, \beta)-$ weak contractive mappings satisfying the following properties:*

 (i) *$TX \subseteq SX$ and $RX \subseteq SX$,*
 (ii) *$T, R$ and $S$ are continuous,*
 (iii) *the pairs $(T, S)$ and $(R, S)$ are compatible,*
 (iv) *$T$ and $R$ are weakly increasing with respect to $S$,*
 (v) *$Sx$ and $Sy$ are comparable.*

*Then $T, R$ and $S$ have a coincidence point $z \in X$.*

**Proof.** Let us assume that $x_0 \in X$ be any arbitrary point in $X$. Since $TX \subseteq SX$ and $RX \subseteq SX$, therefore there exists $x_1, x_2 \in X$ such that $Tx_0 = Sx_1$ and $Rx_1 = Sx_2$. Continuing this way, we can construct sequences $\{x_n\}$ and $\{y_n\}$ in $X$, defined as

$$Sx_{2n+1} = Tx_{2n} = y_{2n}, \quad Sx_{2n+2} = Rx_{2n+1} = y_{2n+1}, \quad \forall n \in N. \tag{3}$$

Since $T$ and $R$ are weakly increasing function with respect to $S$, therefore

$$Sx_1 = Tx_0 \preceq Rx_1 = Sx_2,$$

similarly,

$$Sx_2 = Tx_1 \preceq Rx_2 = Sx_3.$$

Continuing this process, we obtain

$$Sx_1 \preceq Sx_2 \preceq Sx_3......... \preceq Sx_{2n+1} \preceq Sx_{2n+2} \preceq ....$$

Thus,

$$y_0 \preceq y_1 \preceq y_2......... \preceq y_{2n} \preceq y_{2n+1} \preceq ....$$

First we suppose that if there exists $n \in N$ such that $y_{2n-1} = y_{2n}$, then from (1)

$$\begin{aligned}
\psi(d(y_{2n}, y_{2n+1})) &= \psi(d(Tx_{2n}, Rx_{2n+1})) \\
&\leq \alpha(d(Sx_{2n}, Sx_{2n+1}))\beta(d(Sx_{2n}, Sx_{2n+1})) \\
&= \alpha(d(y_{2n-1}, y_{2n}))\beta(d(y_{2n-1}, y_{2n})) = 0,
\end{aligned}$$

which implies that $y_{2n+1} = y_{2n}$. Consequently, $y_m = y_{2n-1}$ for any $m \geq 2n$. Hence for every $m \geq 2n$, we have $Sx_m = Sx_{2n}$. This implies that $\{Sx_n\}$ is a Cauchy sequence.

Secondly, suppose that $y_n \neq y_{n+1}$ for any integer $n$. Let $z_n = d(y_n, y_{n+1})$. Now we show that $z_n \to 0$ as $n \to \infty$.

Since $Sx_{2n}$ and $Sx_{2n+1}$ are comparable, then again from (1) we obtain

$$
\begin{aligned}
\psi(d(y_{2n+2}, y_{2n+1})) = \psi(d(Sx_{2n+3}, Sx_{2n+2})) &= \psi(d(Tx_{2n+2}, Rx_{2n+1})) \\
&\leq \alpha(d(Sx_{2n+2}, Sx_{2n+1}))\beta(d(Sx_{2n+2}, Sx_{2n+1})) \\
&= \alpha(d(y_{2n+1}, y_{2n}))\beta(d(y_{2n+1}, y_{2n})).
\end{aligned}
\tag{4}
$$

By using (2), property of $\psi$ and the fact that $\alpha \in F$, we get

$$
d(y_{2n+2}, y_{2n+1}) \leq d(y_{2n+1}, y_{2n}),
\tag{5}
$$

similarly, we obtain

$$
d(y_{2n+1}, y_{2n}) \leq d(y_{2n}, y_{2n-1}).
\tag{6}
$$

Combining (5) and (6), we have

$$
d(y_{2n+2}, y_{2n+1}) \leq d(y_{2n+1}, y_{2n}) \leq d(y_{2n}, y_{2n-1}).
\tag{7}
$$

It follows that the sequence $\{z_n\}$ is monotonically decreasing, therefore there exists $r \geq 0$ such that

$$
\lim_{n \to \infty} z_n = d(y_n, y_{n+1}) = r.
\tag{8}
$$

Suppose that $r > 0$, then from (4)

$$
\psi(d(y_{2n+2}, y_{2n+1})) \leq \alpha(d(y_{2n+1}, y_{2n}))\beta(d(y_{2n+1}, y_{2n}))
$$

Taking limit as $n \to \infty$, we get $\psi(r) \leq \alpha(r)\beta(r)$. Since $\alpha \in E$ therefore by using (2), we have $\psi(r) < \beta(r) < \psi(r)$. This is a contradiction. Therefore, $r = 0$. Hence

$$
\lim_{n \to \infty} z_n = d(y_n, y_{n+1}) = 0.
\tag{9}
$$

Next, we prove that $\{Sx_n\}$ is a Cauchy sequence. We prove this by negation. Suppose, on the contrary, that $\{Sx_{2n}\}$ is not a Cauchy sequence. Then for any $\epsilon > 0$, there exist two subsequences of positive integers $m_k$ and $n_k$ such that $n_k > m_k > k$ for all positive integer $k$,

$$
d(Sx_{2m_k}, Sx_{2n_k}) > \epsilon \quad and \quad d(Sx_{2m_k}, Sx_{2n_{k-2}}) \leq \epsilon.
\tag{10}
$$

From (10) and by using triangle inequality, we have

$$
\begin{aligned}
\epsilon < d(Sx_{2m_k}, Sx_{2n_k}) \\
\leq d(Sx_{2m_k}, Sx_{2n_{k-2}}) + d(Sx_{2n_{k-2}}, Sx_{2n_{k-1}}) + d(Sx_{2n_{k-1}}, Sx_{2n_k}).
\end{aligned}
$$

Letting $k \to \infty$ in above equality and using (9), we get

$$
\lim_{k \to \infty} d(Sx_{2m_k}, Sx_{2n_k}) = \epsilon.
\tag{11}
$$

Again, by using triangle inequality, we have

$$
d(Sx_{2n_k}, Sx_{2m_{k-1}}) \leq d(Sx_{2n_k}, Sx_{2m_k}) + d(Sx_{2m_k}, Sx_{2m_{k-1}}),
$$

taking limit as $k \to \infty$ in above equality and using (9)–(11), we have

$$\lim_{k\to\infty} d(Sx_{2n_k}, Sx_{2m_{k-1}}) = \epsilon. \tag{12}$$

Since,

$$\begin{aligned}
d(Sx_{2n_k}, Sx_{2m_k}) &\leq d(Sx_{2n_k}, Sx_{2n_{k+1}}) + d(Sx_{2n_{k+1}}, Sx_{2m_k}) \\
&= d(Sx_{2n_k}, Sx_{2n_{k+1}}) + d(Tx_{2n_k}, Rx_{2m_{k-1}}).
\end{aligned}$$

Using (9)–(12) and letting $k \to \infty$, we have

$$\epsilon \leq \lim_{k\to\infty} d(Tx_{2n_k}, Rx_{2m_{k-1}}).$$

However, $\psi \in \Psi$, therefore

$$\psi(\epsilon) \leq \lim_{k\to\infty} \psi(d(Tx_{2n_k}, Rx_{2m_{k-1}}))). \tag{13}$$

From (1), we have

$$\psi(d(Tx_{2n_k}, Rx_{2m_{k-1}})) \leq \alpha(d(Sx_{2n_k}, Sx_{2m_{k-1}}))\beta(d(Sx_{2n_k}, Sx_{2m_{k-1}})).$$

Taking limit $k \to \infty$ in above inequality and using the fact that $\alpha \in F$, we get

$$\lim_{k\to\infty} \psi(d(Tx_{2n_k}, Rx_{2m_{k-1}}))) < \beta(\epsilon). \tag{14}$$

From (13) and (14) and using (2), we get

$$\psi(\epsilon) \leq \lim_{k\to\infty} \psi(d(Tx_{2n_k}, Rx_{2m_{k-1}}))) < \beta(\epsilon) < \psi(\epsilon). \tag{15}$$

This is a contradiction. Therefore $\{Sx_{2n}\}$ is a Cauchy sequence and hence $\{Sx_n\}$ is a Cauchy sequence for all $n$. Hence there exist $u \in X$ such that

$$\lim_{n\to\infty} Sx_n = u. \tag{16}$$

Next, we claim that $u$ is a coincidence point of $T$, $R$, and $S$.
From (16) and the continuity of $S$, we get

$$\lim_{n\to\infty} S(Sx_n) = Su. \tag{17}$$

From triangular inequality, we have

$$d(Su, Tu) \leq d(Su, S(Sx_{2n+1})) + d(S(Tx_{2n}), T(Sx_{2n})) + d(T(Sx_{2n}), Tu), \tag{18}$$

From (3) and (16), we have

$$Sx_{2n} \to u, \qquad Tx_{2n} \to u. \tag{19}$$

Since pair $(T, S)$ is compatible, then

$$d(S(Tx_{2n}), T(Sx_{2n})) \to 0. \tag{20}$$

Using the continuity of $T$ and (19), we have

$$d(T(Sx_{2n}), Tu) \to 0. \tag{21}$$

Letting $k \to \infty$ in (18) and using (17)–(20) together with (21), we get

$$d(Su, Tu) \leq 0,$$

which means that $Su = Tu$.

Similarly from triangular inequality, we have

$$d(Su, Ru) \leq d(Su, S(Sx_{2n+2})) + d(S(Rx_{2n+1}), R(Sx_{2n+1})) + d(R(Sx_{2n+1}), Ru), \tag{22}$$

In similar manner, we get $d(Su, Ru) \leq 0$, which means that $Su = Ru$. Thus, we find that $Su = Tu = Ru$, that is, $u$ is a coincidence point of $T$, $R$, and $S$. This proves Theorem 10. $\square$

Now we give a sufficient condition for the uniqueness of the common fixed point in Theorem 10. This condition is as follows:

$$for\ (x, y) \in X \times X,\ there\ exists\ u \in X\ such\ that\ Tx \preceq Tu\ and\ Ty \preceq Tu. \tag{23}$$

**Theorem 11.** *Adding the condition (23) to the hypotheses of Theorem 10, the self maps $T$, $R$, and $S$ have a unique common fixed point .*

**Proof.** First, we prove that $T$, $R$, and $S$ have common fixed point. To prove this, we show that if $p$ and $q$ are coincidence points of $T$, $R$, and $S$, i.e.,

$$Sp = Tp = Rp \quad and \quad Sq = Tq = Rq$$

then

$$Sp = Sq. \tag{24}$$

From our assumption, there exists $u_0 \in X$ such that

$$Tp \preceq Tu_0, \quad Tq \preceq Tu_0. \tag{25}$$

Now we follow the proof of Theorem 10, we can define a sequence $\{Su_n\}$ as follows:

$$Su_{2n+1} = Tu_{2n}, \quad Su_{2n+2} = Ru_{2n+1}, \quad \forall n \in N. \tag{26}$$

Again, we have

$$Tp = Sp \preceq Su_n, \quad Tq = Sq \preceq Su_n, \quad \forall n \in N. \tag{27}$$

Now put $x = u_{2n}$ and $y = p$ in (1), we get

$$\psi(d(Su_{2n+1}, Sp)) = \psi(d(Tu_{2n}, Rp))$$
$$\leq \alpha(d(Su_{2n}, Sp))\beta(d(Su_{2n}, Sp)).$$

Since $\alpha \in F$,

$$\psi(d(Su_{2n+1}, Sp)) \leq \beta(d(Su_{2n}, Sp)). \tag{28}$$

Similarly, again if we put $y = u_{2n}$ and $x = p$ in (1), we obtain

$$\psi(d(Su_{2n+2}, Sp)) \leq \beta(d(Su_{2n+1}, Sp)). \tag{29}$$

Combine (28) and (29) for all $n \in N$, we get

$$\psi(d(Su_{n+1}, Sp)) \leq \beta(d(Su_n, Sp)),$$

Consequently, by using property of $\psi$ and $\beta$

$$d(Su_{n+1}, Sp) \leq d(Su_n, Sp),$$

therefore, there exists $r \geq 0$ such that

$$\lim_{n \to \infty} d(Su_n, Sp) = r. \tag{30}$$

Suppose $r > 0$, then from (1)

$$\psi(d(Su_{2n+1}, Sp)) \leq \alpha(d(Su_{2n}, Sp))\beta(d(Su_{2n}, Sp)),$$

on taking limit as $n \to \infty$ and using (2), we get

$$\psi(r) < \beta(r) < \psi(r).$$

This is a contradiction. Thus, $r = 0$, therefore from (30), we have

$$\lim_{n \to \infty} d(Su_n, Sp) = 0. \tag{31}$$

In same manner, we can show that

$$\lim_{n \to \infty} d(Su_n, Sq) = 0. \tag{32}$$

Now using the fact that limit is unique and by using (24)–(32), we can write

$$\lim_{n \to \infty} Tu_{2n} = Sp = Sq, \quad \lim_{n \to \infty} Ru_{2n+1} = Sp = Sq. \tag{33}$$

Since the pair $\{T, S\}$ and $\{R, S\}$ are compatible, therefore

$$\lim_{n \to \infty} d(S(Tu_{2n}), T(Su_{2n})) = 0, \quad \lim_{n \to \infty} d(S(Ru_{2n+1}), R(Su_{2n} + 1)) = 0. \tag{34}$$

Let us take,

$$z = Sp \tag{35}$$

Consider,

$$d(Sz, Tz) \leq d(Sz, S(Tu_{2n})) + d(S(Tu_{2n}), T(Su_{2n})) + d(T(Su_{2n}), Tz).$$

Letting $n \to \infty$ and using the continuity of $T$ as the above inequality, we get

$$d(Sz, Tz) \leq 0,$$

that is, $Sz = Tz$ and $z$ is the coincidence point of $T$ and $S$.

Similarly, proceeding as above, we can write

$$d(Sz, Rz) \leq 0,$$

that is, $Sz = Rz$ and $z$ is the coincidence point of $R$ and $S$.

Hence, from (24), we have

$$z = Sp = Sz = Tz = Rz.$$

This proves that $z$ is a common fixed point of $T, R$, and $S$.

**Uniqueness:** Next we prove that the common fixed point is unique. Assume that the fixed point is not unique, therefore there exists another fixed point $\lambda \in X$ such that

$$\lambda = Sp = S\lambda = T\lambda = R\lambda.$$

Using (24), we have

$$S\lambda = Sz.$$

Hence we get,

$$\lambda = S\lambda = Sz = z,$$

this is a contradiction to our assumption and hence common fixed point is unique. This completes the proof of the Theorem 11. □

If we take $S = I$ in Definition 9 and Theorem 10, we get the following result.

**Definition 10.** *Two self maps $T, R$ are said to be $(\psi, \alpha, \beta)-$ weak contraction if for each $x, y \in X$*

$$\psi(d(Tx, Ry)) \leq \alpha(d(x, y))\beta(d(x, y)), \quad \forall x \geq y, \tag{36}$$

*where $\alpha \in F$, $\psi \in \Psi$ and $\beta : [0, \infty) \to [0, \infty)$ is a continuous function with condition*

$$0 < \beta(t) < \psi(t), \forall t > 0. \tag{37}$$

**Theorem 12.** *Let $(X, \preceq)$ be a partially ordered set and suppose that there exists a metric $d$ in $X$ such that $(X, d)$ is a complete metric space. Let $T, R : X \to X$ are $(\psi, \alpha, \beta)-$ weak contractive mappings satisfying the following properties:*

*(i) T and R are continuous,*
*(ii) T and R are weakly increasing,*
*(iii) x and y are comparable.*

*Suppose, if*

$$for\ (x, y) \in X \times X,\ there\ exists\ u \in X\ which\ is\ comparable\ to\ x\ and\ y.$$

*Then T and R have unique common fixed point $z \in X$.*

An example of $(\psi, \alpha, \beta)-$ weak contraction is as follows:

**Example 2.** *Let $X = [0,1]$ and $d(x,y) = \max\{x,y\}$ for all $x,y \in X$. Let $T, R : X \to X$ be defined by*

$$Tx = \begin{cases} \frac{x}{3}, & \text{if } x \neq 1 \\ \frac{1}{6}, & \text{if } x = 1. \end{cases}$$

*and*

$$Rx = \begin{cases} \frac{x}{2}, & \text{if } x \neq 1 \\ \frac{1}{4}, & \text{if } x = 1. \end{cases}$$

*Then $(X, d)$ is a complete metric space.*

*Define maps $\psi, \beta : [0, \infty) \to [0, \infty)$ and $\alpha : [0, \infty) \to [0, 1)$ as $\psi(t) = 2t$, $\beta(t) = t$ and $\alpha(t) = \frac{1}{10}$. Then, all condition $(i), (ii), (iii)$ are satisfied Theorem 12. Hence T and R have unique common fixed point.*

## 3. Applications

In application, here we give an existence theorem for common solutions of integral equations. However, the existence and uniqueness conditions obtained here are weaker than those in the previous studies.

*A Common Solution of Integral Equations by Existence Theorem*

The purpose of this section is to give an example of integral equations, where we can apply Theorem 12 to get common solutions. The following example is motivated by [12,14].

We consider the integral operator

$$\int_0^L K_1(t, s, u(s))ds + \lambda(t)$$

$$\int_0^L K_2(t, s, u(s))ds + \lambda(t) \quad \forall\, t \in [0, L], \tag{38}$$

where $L > 0$. Let us consider the space $X = C(I)$ $(I = [0, L])$ of the continuous functions defined on $I$. Obviously, this space with the metric given by:

$$d(x, y) = \sup_{t \in I} |x(t) - y(t)|, \quad \forall\, x, y \in X,$$

is a complete metric space. $X = C(I)$ can also be prepared with partial order $\preceq$ given by:

$$\forall\, x, y \in X, x \preceq y \Leftrightarrow x(t) \leq y(t), \quad \forall\, t \in I.$$

**Theorem 13.** *Suppose the following hypotheses hold:*

*(i)  $K_1, K_2 : I \times I \times R \to R$ and $h : R \to R$ are continuous,*
*(ii)  for all $t, s \in I$, we have*

$$K_1(t, s, u(t)) \leq K_2\left(t, s, \int_0^L K_1(s, z, u(z))dz + h(s)\right),$$

$$K_2(t, s, u(t)) \leq K_1\left(t, s, \int_0^L K_2(s, z, u(z))dz + h(s)\right),$$

(iii)   *there exists a continuous function* $G : I \times I \to R_+$ *such that*

$$|K_1(t, s, x) - K_2(t, s, y)| \leq G(t, s)\sqrt{\frac{log[(x-y)^2 + 1]}{(x-y)}},$$

$$\forall \, t, s \in I \text{ and } x, y \in R \text{ such that } y \preceq x,$$

(iv)   $\sup_{t \in I} \int_0^L G^2(t, s)ds \leq \frac{1}{L}.$

*Then the integral Equation (38) have a solution* $u^* \in C(I)$.

**Proof.** Let us define $T, R : C(I) \to C(I)$ by :

$$Tx(t) = \int_0^L K_1(t, s, u(s))ds + h(t)$$

and

$$Rx(t) = \int_0^L K_2(t, s, u(s))ds + h(t), \quad t \in I, \ x \in C(I).$$

Nashine and Samet in [18] showed that $T$ and $R$ are weakly increasing. Now, for all $x, y \in C(I)$ such that $y \preceq x$, we have:

$$|Tx(t) - Ry(t)| \leq \int_0^L |K_1(t, s, x(s)) - K_2(t, s, y(s))| \, ds$$

$$\leq \int_0^L G(t, s)\sqrt{\frac{log[(x(s) - y(s))^2 + 1]}{(x(s) - y(s))}} ds. \tag{39}$$

Using Cauchy-Schwarz inequality in the R.H.S. of (39), we get

$$\int_0^L G(t, s)\sqrt{\frac{log[(x(s) - y(s))^2 + 1]}{(x(s) - y(s))}} ds \leq \left(\int_0^L G^2(t, s)ds\right)^{\frac{1}{2}} \left(\int_0^L \frac{log[(x(s) - y(s))^2 + 1]}{(x(s) - y(s))} ds\right)^{\frac{1}{2}}.$$

Now using hypothesis(IV), we get

$$\int_0^L G(t, s)\sqrt{\frac{log[(x(s) - y(s))^2 + 1]}{(x(s) - y(s))}} ds \leq \left(\frac{1}{L}\right)^{\frac{1}{2}} \left(\int_0^L \frac{log[(x(s) - y(s))^2 + 1]}{(x(s) - y(s))} ds\right)^{\frac{1}{2}}$$

$$\leq \left(\frac{1}{L}\right)^{\frac{1}{2}} \left(\sqrt{\frac{log[d(x,y)^2 + 1]}{d(x,y)}}\right)\sqrt{L}$$

$$\leq \left(\sqrt{\frac{log[d(x,y)^2 + 1]}{d(x,y)}}\right).$$

Hence from (39),

$$|Tx(t) - Ry(t)| \leq \sqrt{\frac{log[d(x,y)^2 + 1]}{d(x,y)}}. \tag{40}$$

This implies that

$$d(Tx, Ry) \leq \sqrt{\frac{log[d(x,y)^2 + 1]}{d(x,y)}}$$

$$d(Tx, Ry)^2 \leq \frac{log[d(x,y)^2 + 1]}{d(x,y)} \leq \frac{\sqrt{log[d(x,y)^2 + 1]}}{d(x,y)} \cdot \sqrt{log[d(x,y)^2 + 1]}. \qquad (41)$$

Let us choose a function $\alpha$ as,

$$\alpha(t) = \frac{\sqrt{log[t^2 + 1]}}{t},$$

it is clear that with this choice, $\alpha \in S$. Also assume that $\psi(t) = t^2$ and $\beta(t) = \sqrt{log[t^2 + 1]}$.

Therefore from (41), we have

$$\psi(d(Tx, Ry)) \leq \alpha(d(x,y))\beta(d(x,y)).$$

Since all the hypotheses of Theorem 12 are satisfied. Therefore, there exists $u^* \in C(I)$, a common fixed point of $T$ and $R$, that is, $u^*$ is a solution to (38). $\square$

**Author Contributions:** All authors have equal contribution in this paper.

**Funding:** This project was supported by the Rajamangala University of Technology Thanyaburi (RMUTTT) (Grant No. NSF62D0604).

**Acknowledgments:** P.K. would like to thank the Theoretical and Computational Science (TaCS) Center under Computational and Applied Science for Smart Innovation research Cluster (CLASSIC), Faculty of Science, KMUTT.

**Conflicts of Interest:** The authors declare no conflict of interest.

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
