# Peer review of "Generalized (ψ,α,β)—Weak Contractions for Initial Value Problems"

_mathematics, doi:10.3390/math7030266_

Round 1

Reviewer 1 Report

My comments are in the attached file

Author Response

Dear Reviewer 1

We revised all the problems suggested by Reviewer 1. 

(1) It seems to me that the sentence: ”Since T and R are weakly increasing function with respect to S” page 8, line 4 is unclear. ”Definition 1.7 states thatT;Rareweaklyincreasingif...... nothingissaidabout”withrespect to” If we do not mind this misprint (which should be corrected) in my opinion in Definition 1.7 the following inequality is assumed. T x ≼ RT x But in page 8,line 5 the inequality is Tx0 ≼ Rx1 The point x1 is defined in page 7, last line as Tx0 = Sx1, i.e. x1 = S−1(Tx0). I could not see how we can apply de nition 1.7. • Revised: We have corrected it for accuracy and Because T : X → X and (X,d) is acomplete metric space, Tx0 = x1 and Tx1 = x2, so Tx0 ≼ RTx0 =Rx1. 

(2) Example 1. The maps T ; R; S : N ∪ {0} → Q+. In Definition 2.1 nothing is said about the maps, except that thy are defined for any x ∈ X, but where is their image seems unclear. Theorem 10 considerer maps T ; R; S : X → X. That is why in my opinion Example 1 should be an example of maps that satisfy theorem 10. • Revised: Example 1 is example of definition 2.1 which we wrong type from a generalized(φ, α, β) − contraction → a generalized(φ, α, β) − weak contraction. 

(3) In the application section in (38) page 16 an integral operator is written, but the authors are saying that integral equations are written. This is a misprint in my opinion. • Revised: We modified as suggested. 

(4) A solution of a system of integral inequalities is presented. Once proof is corrected it will be good to present a particular choice of such a system, because it may happen that the considered system is an empty set.

Revised: In theorem 13, we mentioned a solution of integral equation. Next time, we may apply to a solution of system integral.

Thank you very much for valuable important suggestions of Reviewers 1

and Reviewers 2 to improve our paper and recommend for publication. 

Sincerely yours,

Piyachat Borisut Poom Kumam Vishal Gupta Naveen Mani

Reviewer 2 Report

The paper contains  some interesting  new  formal results.  Some suggestions to be take into account in a revised version follow below:

Statement of Theorem 8, Theorem 6  and  line before Theorem 8 : The order relation symbol  is sometimes written as  and sometimes as . Revise this notation thoroughly.

Why the proofs of Theorems 10 , 11 are named  as Proof 1, Proof 2 ? and Theorem 13, proof 3?.

The main body in section 2 has only a simple example. More  nontrivial examples  have to be incorporated to illustrate the main results,.

There are many articles like “ a” or “ the”  which are missed in many

sentences of the text. Some examples follow:

Line 3 after Definition 1.8 “ in the sense”

Line 2 before section 2 “ a common solution”

Page 6, Def.2. “ to be a generalized…”

Page 12, line 2, a unique

Page 14, uniqueness : common fixed point-> the common fixed point

Page 14, line 3 from the bottom: same correction as above.

Page 14, line 7 : the coincidence point

Page 14, line 3, the above inequality

Revise these points thoroughly.

 Some related background literature is recommended  for its inclusion with some brief  related comments to add in the introduction as follows:

A common  fixed point theorem for generalized ( psi, phi)-weak contractions of Suzuki type,  Journal of Mathematical Analysis, Vol. 8, Issue 2, pp. 80-88, 2017.

A fixed point theorem for generalized weak contractions, Filomat, Vol. 29 , Issue 7, pp. 1481-1490, 2015.

Author Response

Dear Reviewer 2

We revised all the problems suggested by Reviewer 2.

(1) Statement of Theorem 8, Theorem 6 and line before Theorem 8 : The order relation symbol is sometimes written as and sometimes as . Revise this notation thoroughly.

Revised: We modified as suggested.
(2) Why the proofs of Theorems 10 , 11 are named as Proof 1, Proof 2 ? and

Theorem 13, proof 3 ?.
Revised: We wrong type from proof 1 proof, proof 2 proof, proof 3

proof.
(3) The main body in section 2 has only a simple example. More nontrivial

examples have to be incorporated to illustrate the main results.Revised: We have added example in section 2.

(4) There are many articles like a or the which are missed in many sentences of the text.

Revised: We modified as suggested.

(5) Some related background literature is recommended for its inclusion with some brief related comments to add in the introduction as follows:
A common fixed point theorem for generalized ( psi, phi) weak contractions of Suzuki type, Journal of Mathematical Analysis, Vol. 8, Issue 2, pp. 8088, 2017.

A fixed point theorem for generalized weak contractions, Filomat, Vol. 29, Issue 7, pp. 14811490, 2015

Revised: We have added relation commonts in the introduction.

Thank you very much for valuable important suggestions of
Reviewers 1

and Reviewers 2 to improve our paper and recommend for publication. 

Sincerely yours,

Piyachat Borisut Poom Kumam Vishal Gupta Naveen Mani

Round 2

Reviewer 1 Report

I am satisfied with the revisions made by the authors.

I recommend the manuscript to be accepted for a publication in the present form.

Reviewer 2 Report

The paper is welll- written, well- formalized and well-organized. The proofs are addressed with clarity and rigor and the subject agrees with the journal scope.